# Unraveling Max-Return Sequence Modeling via Return Consistency

## Abstract

Offline reinforcement learning (RL) learns from fixed datasets without interaction with online environment, enabling supervised solutions for offline RL. Decision Transformer (DT) casts offline RL as return-conditioned supervised sequence modeling, thereby sidestepping optimal value fitting and policy gradients. This paradigm overlooks RL's core objective of return maximization, which yields brittle behavior on suboptimal trajectories and limited stitching ability. Rein*for*mer reorients this objective through max-return sequence modeling: during inference, the model conditions on the predicted maximum achievable returns to generate the optimal actions. To better understand both the SOTA performance of this paradigm and its occasional dramatic failures, we adopt a supervised perspective and introduce the **return consistency** to assess whether similar state-action pairs have similar returns. Indeed, high return consistency guarantees the maximized return reliably cues the optimal action, while low consistency may lead to suboptimal action selection. Through visualizations, two different consistency modes are exposed and we quantify this via the return standard deviation of the data cluster with highest return mean. Furthermore, we reveal the relationship between this metric and 1) final performance, 2) context lengths, 3) model architectures through a systematic study. Finally, we improve return consistency by explicitly decreasing the return standard deviation, thereby further increasing the performance.

## 1 Introduction

Classical online reinforcement learning (RL) algorithms such as Q-learning (Watkins & Dayan, 1992) or policy gradient (Sutton et al., 1999) are derived from the Markov Decision Process (MDP) (Sutton et al., 1998) formulation, a paradigm fundamentally different from supervised learning. Offline RL (Levine et al., 2020; Fu et al., 2020), learning from static datasets rather than dynamic environment, takes a step closer to the supervised paradigm. On the basis, sequence modeling (Chen et al., 2021) maximizes the likelihood of actions based on the whole historical trajectories that including state, action and returns. In this way, offline RL is addressed from one paradigm similar to the supervised learning. A particularly enticing prospect is that the successes of supervised sequence modeling in other domains (Dosovitskiy et al., 2020; Ouyang et al., 2022) may be replicable within the offline realm, potentially catapulting the rapid advancement and progress of reinforcement learning.

Sequence modeling, also Decision transformer (DT) (Chen et al., 2021), has two significant limitations. One is the manual specification of the initial target return during inference, which typically necessitates expert knowledge or extensive experiments to determine (Zheng et al., 2022). The other is that this approach completely abandons the core objective of RL—maximizing return, causing poor trajectory stitching ability (Brandfonbrener et al., 2022). Consequently, Reinforced Transformer (Rein*for*mer) (Zhuang et al., 2024) introduces the concept of max-return sequence modeling, which brings the return maximization back to this supervised paradigm. Rein*for*mer predicts the maximized return to guide the generation of actions during inference, eliminating the need to specify an initial target return and also enhancing the stitching ability compared with basic sequence modeling.

Moreover, when and why max-return sequence modeling outperforms or underperforms conventional offline RL remains an open question, stalling further improvement. To address this, we conduct an in-depth investigation into the inference stage from the data perspective. Intuitively, similar state-action pairs should carry similar returns; only under this condition does maximizing predicted

return guarantee selecting the truly optimal action. If the assumption is violated, the model may blindly pick actions whose nearest neighbors map to low returns, triggering an abrupt performance collapse. We refer to the scenario where similar state-action pairs yield similar returns as **high return consistency**. Indeed, visualizations across different datasets expose two distinct return-consistency regimes and we quantify this via *the return std of the cluster with highest return mean*.

Surrounding the return consistency, we explore the relationships between this concept and 1) the performance, 2) context length, as well as 3) model architecture. In summarize, datasets with higher consistency tend to exhibit superior performance for max-return sequence modeling, which also favors longer context length and benefit more from architectures that focus on global information. Furthermore, the performance of max-return sequence modeling is clear to improve. Enhancing return consistency, also reducing the standard deviation of returns, exactly suggests that classical variance-reduction technique from traditional RL is helpful. Specifically, we substitute raw returns with advantages, the expectation of return minus the value baseline. This replacement substantially improves return consistency and leads to significant gains in final performance. This phenomenon, achieved with only return replacement, constitutes indirect validation of the return consistency itself.

## 2 PRELIMINARY

### 2.1 OFFLINE REINFORCEMENT LEARNING

Offline RL (Levine et al., 2020) forbids the interaction with the environment and only a fixed offline dataset full of trajectories $\mathcal{D} = \{(s_0, a_0, r_0, s_1, a_1, r_1, \cdots, s_t, a_t, r_t \cdots)\}$ is provided . Here $s_t$ is the current state at timestep $t$, $a_t$ is the action and $r_t \dot{=} r(s_t, a_t)$ is the reward of current state and action. The objective of offline RL is to learn a policy $\pi(a_t|s_t)$ that maximizes the expected returns $\mathbb{E}_\pi\left[\sum_{t=0}^T r(s_t, a_t)\right]$. Compared to the traditional online RL (Sutton et al., 1998), this setting is more challenging since the agent is unable to explore the environment and collect extra feedback.

### 2.2 SEQUENCE MODELING

Sequence modeling (Chen et al., 2021) breaks the traditional Markov property, where the prediction of the current action $a_t$ is based on the previous $K$ (also called context length) timesteps trajectories $\tau_{t-K} = (R_{t-K+1}, s_{t-K+1}, a_{t-K+1}, \cdots, R_{t+1}, s_{t+1}, a_{t+1})$ where $R_t \dot{=} \sum_{t'=t}^T r_{t'}$ represents the sum of future rewards from the current timestep $t$, known as returns-to-go (or simply returns).

Sequence modeling directly maximizes the likelihood of actions conditioned on not only the current state $s_t$ and returns-to-go $R_t$, but also the historical trajectories $\tau_{t-K}$:

$$\mathcal{L}_{\text{DT}} = -\mathbb{E}_t\left[\log \pi(a_t|\tau_{t-K}, s_t, R_t)\right]. \tag{1}$$

This loss indicates offline RL is solved from the supervised perspective, rather than traditional RL paradigm. Besides, $\pi$ is implemented based on sequence model transformer (Vaswani et al., 2017).

For the **Inference**, the initial target returns $\hat{R}_0$ must be determined first. Given $\hat{R}_0$ and the initial environment state $s_0$, the next action is generated by the model $\pi\left(a_1|\hat{R}_0, s_0\right)$. Once the action $a_1$ is executed, the environment returns the next state $s_1$ and reward $r_1$ are returned. The next return-to-go is updated as $\hat{R}_1 = \hat{R}_0 - r_1$. This process is repeated until the episode terminates.

### 2.3 MAX-RETURN SEQUENCE MODELING

Since supervised sequence modeling does not explicitly consider return maximization, the core objective of RL, the concept of max-return sequence modeling is introduced. The key lies in utilizing the maximized return to guide the generation of next actions during inference. Concretely, Reinforced Transformer (Rein*for*mer) (Zhuang et al., 2024) adopt the following historical trajectories $\tau_{t-K} = (s_{t-K+1}, R_{t-K+1}, a_{t-K+1}, \cdots, s_{t+1}, R_{t+1}, a_{t+1})$ where the state $s_t$ is placed before the returns-to-go $\hat{R}_t$, different from the original DT. The main advantage is that the return can be predicted through the state without the need for prior specification. During training, Rein*for*mer introduces a

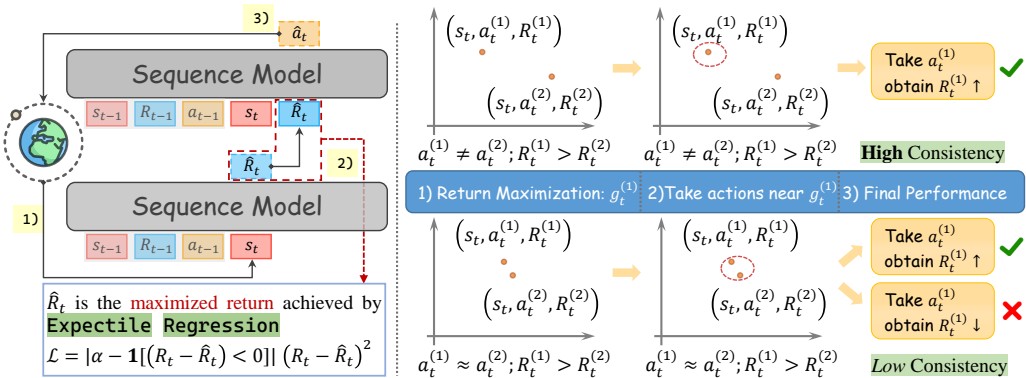

Figure 1: **Left:** Max-return sequence modeling, during inference, first predicts the maximized return using expectile regression. And then this predicted maximized return is reintroduced back into the same transformer to guide the optimal action generation. **Right:** The visualization of two possible action selection process under **high** or *Low* return consistency scenarios.

return loss based on expectile regression in addition to the action probability maximization loss:

$$\mathcal{L}_{\text{Reinformer}} = -\mathbb{E}_t \log \pi \left( a_t | \tau_{t-K}, s_t, R_t \right) + \mathbb{E}_t \left[ |\alpha - \mathbb{1} \left( \Delta R_t < 0 \right)| \, \Delta R_t^2 \right],$$

where $\Delta R_t = R_t - \pi \left( \hat{R}_t | \tau_{t-K}, s_t \right)$ is the difference between the oracle return $R_t$ and its prediction $\hat{R}_t$. Here $\alpha \in (0, 1)$ is the hyperparameter of expectile regression. When $\alpha = 0.5$, expectile regression degenerates into standard MSE loss. But when $\alpha > 0.5$, this asymmetric loss gives more weight to $R_t$ larger than $\hat{R}_t$. It can be proved that this additional return loss can make the model predict the maximum returns-to-go when $\alpha \to 1$, similar to the return maximization objective in RL.

When **inference**, given the initial state $s_0$, the maximized initial target return is predicted $\pi \left( \hat{R}_0 | s_0 \right)$ rather than manually specified. Since $\hat{R}_0$ is maximized, the next action $\pi \left( a_1 | \hat{R}_0, s_0 \right)$ will approach to the optimal one. The next state $s_1$ is then returned and this process is repeated until termination.

## 3 RETURN CONSISTENCY

We first conduct a detailed analysis of the potential issues during the inference phase of max-return sequence modeling. Based on this, the concept of high return consistency is introduced. We also develop a concretely metric to measure return consistency with the help of clustering. In addition, we explore the relationships between return consistency and the performance, context length, model architecture, drawing several conjectures. Finally, we enhance return consistency by deploying the classical RL variance-reduction trick, which is strikingly simple yet measurably strong.

**a) Analysis of Inference Stage**  At timestep $t$, the max-return sequence modeling employs the following inputs and outputs during training:

$$\textbf{Input:} \ \left\langle \tau_{t-K}, s_t, R_t \right\rangle \xrightarrow{\pi} \textbf{Output:} \ \left\langle \hat{R}_t, \hat{a}_t \right\rangle. \tag{2}$$

The model attempts to establish a connection between return $R_t$ and action $a_t$. During the inference phase, max-return sequence modeling first predicts a maximized return $\hat{R}_t^{\max}$ and then attempts to forecast the optimal action $\hat{a}_t^*$ under the guidance of this maximized return:

$$\textbf{Input:} \ \left\langle \tau_{t-K}, s_t, \hat{R}_t^{\max} \right\rangle \xrightarrow{\pi} \textbf{Output:} \ \left\langle \hat{a}_t^* \right\rangle. \tag{3}$$

We illustrate the principle and potential issues of this inference with the example in right part of Figure 1. Assume we have two datapoints $\left( \tau_{t-K}, s_t, a_t^{(1)}, R_t^{(1)} \right)$ and $\left( \tau_{t-K}, s_t, a_t^{(2)}, R_t^{(2)} \right)$ where $R_t^{(1)} \geq R_t^{(2)}$. During the inference phase, two scenarios may occur:

- **Problematic Case** $a_t^{(1)} \approx a_t^{(2)}$: But when $a_t^{(1)}, a_t^{(2)}$ are similar, the process of predicting $a_t^{(1)}$ may erroneously yield $a_t^{(2)}$, leading to a decline in performance.

- **Ideal Case** $a_t^{(1)} \neq a_t^{(2)}$: Max-return sequence modeling first predicts a value close to $R_t^{(1)}$ as the maximized return. Since $a_t^{(1)}, a_t^{(2)}$ are not similar, the model $\pi$ will accurately predict $a_t^{(1)}$ under the guidance of $R_t^{(1)}$, thereby maximizing the return.

The above **Ideal Case** can be rigorously formulated using the following definition:

**Definition 3.1.** *Given a dataset $D = \{(s_t, a_t, R_t)\}_{t=1}^N$ consisting of state-action-return triplets, where $(s_t, a_t)$ is a state-action pair, $R_t$ is the corresponding return, $d_{sa} : (\mathcal{S} \times \mathcal{A}) \times (\mathcal{S} \times \mathcal{A}) \to \mathbb{R}^+$ is a distance metric in the state-action space, $d_g : \mathcal{G} \times \mathcal{G} \to \mathbb{R}^+$ is the Euclidean distance. The dataset belongs to **Ideal Case** if the following holds:*

$$\forall \varepsilon > 0, \ \exists \delta > 0, \ s.t. \ \sup_{t \in [1,N]} \max_{t' \in B_\delta(t)} d_g(R_t, R_{t'}) < \varepsilon, \tag{4}$$

*where the neighborhood $B_\delta(t) = \{t' \in [1, N] \mid d_{sa}[(s_t, a_t), (s_{t'}, a_{t'})] < \delta\}$.*

Simply put, this definition implies that when state-action pairs are sufficiently close to each other, their corresponding returns are also close. Under such scenarios, the max-return sequence modeling can distinguish the quality of different actions. Therefore, we refer to datasets that satisfy this definition as having **high return consistency**, and those that do not as having *low return consistency*.

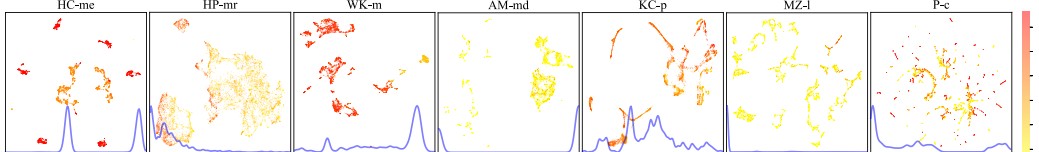

Figure 2: The state-action distribution, obtained through dimensionality reduction using Umap (McInnes et al., 2018), is visualized alongside the KDE curves (blue) of the returns. `AM-mp` and `P-h` are omitted here due to the similarity with `AM-md` and `P-c`.

**b) Visualization and Metric** In this paper, we consider 9 representative datasets including 1) common `Halfcheetah-medium-expert`, `Hopper-medium-replay`, `Walker2d-medium`, 2) trajectory stitching domain `Antmaze-medium-play`, `Antmaze-medium-diverse` and 3) more diverse `Kitchen-partial`, `maze2d-large`, `Pen-human` and `Pen-cloned` with various characteristic[1]. For the rationale behind the selection of these datasets, please refer to Appendix C. To verify whether these datasets meet the definition of **high** return consistency, we visualize the state-action distributions in Figure 2, with colors indicating the magnitude of the returns.

In Figure 2, we can observe that for `HC-me` (the first one), the outer ring consists entirely of high returns, while the inner ring comprises medium returns, with a very clear boundary. This aligns with the definition of **high** return consistency. In contrast, for `AT-md` (the fourth one), sporadic red dots are interspersed among the yellow dots, indicating a clear presence of state-action pairs that are similar yet have significantly different returns. So `AT-md` is *low* return consistency.

Table 1: The return consistency defined by the return std of the cluster with highest return mean.

| Datasets | HC-me | HP-mr | WK-m | AT-md | KC-p | MZ-l | P-c |
|---|---|---|---|---|---|---|---|
| Return Range | $[-2.67, 1.22]$ | $[-1.25, 3.00]$ | $[-3.29, 1.20]$ | $[-0.21, 4.73]$ | $[-2.45, 2.46]$ | $[-0.32, 5.67]$ | $[-0.96, 1.73]$ |
| Return Mean | 1.07 | 2.05 | 0.58 | 0.06 | 1.88 | 3.16 | 1.73 |
| Return Std | 0.03 | 0.94 | 0.05 | 1.12 | 0.07 | 1.00 | 0.00 |
| *consistency* | **High** | *Low* | **High** | *Low* | **High** | *Low* | **High** |

Inspired by the above visualization, we further develop a metric for return consistency across different datasets. DBSCAN (Ester et al., 1996) is first applied to cluster state-action pairs. Then, for each cluster, we compute the mean return (to identify high-return regions) and standard deviation of returns (to measure variability). We use the std of the highest-mean-return cluster as the metric for return consistency, since 1) RL focus on high-return regions and 2) std quantifies the spread of returns—smaller values indicate better consistency. Since the returns are normalized, allowing us to define return consistency as follows:

$$\textbf{High return consistency}: \text{std} \approx 0; \quad \textit{Low return consistency}: \text{std} \approx 1. \tag{5}$$

---

[1]These datasets are abbreviated as `HC-me`, `HP-mr`, `WK-m`, `AM-mp`, `AM-md`, `K-p`, `MZ-l`, `P-h`, `P-c`.

**c) The relationship between return consistency and performance, context length, model architecture.** In datapoint $\left(\tau_{t-K}, s_t, a_t^{(1)}, R_t^{(1)}\right)$, the context length affects $\tau_{t-K}$. The longer the context length $K$, the more informative $\tau_{t-K}$ is, thereby making the following mapping more robust:

$$\left\langle \tau_{t-K}, s_t, \hat{R}_t^{(1)} \right\rangle \stackrel{\pi}{\longrightarrow} \left\langle \hat{a}_t^{(1)} \right\rangle. \tag{6}$$

Additionally, when the data exhibit high consistency, the validity of this mapping positively impacts performance. Therefore, the higher return consistency and the longer context length, the better performance. We consider context length $K = 2, 5, 10, 20$, with the maximum value of 20 being the default of DT (Chen et al., 2021), and the minimum of 2 corresponding to the shortest length.

Moreover, when the return consistency is high and the context length $K$ is long, model architecture that can capture global information gain a significant advantage. In our experiments, we investigated the impact of three model architectures on the final performance, including the Transformer (Vaswani et al., 2017), One-dimensional convolutional layers (Conv) (Yu et al., 2022; Kim et al., 2023), and the linear Recurrent Neural Network Mamba (Gu & Dao, 2023; Cao et al., 2024; Huang et al., 2024; Ota, 2024; Lv et al., 2024). Their corresponding sequence models are called Rein*for*mer, Reinconver and Reimba (see Appendix D). Since the Transformer itself lacks the ability to understand relative positional relationships, we augmented it with positional encoding, while the other two models not.

---

**Conjectures**

1) The performance of max-return sequence modeling is positively correlated with **return consistency**. 2) **High** consistency datasets prefer **longer** context length $K$. 3) Architecture that excels at understanding **global** information are more compatible with **high** consistency.

---

**d) return consistency with advantage** Previous first conjecture about performance is verified in section 5.1, motivating a simple yet effective upgrade to max-return sequence modeling: increasing return consistency by decreasing the standard deviation of returns inside the highest-mean-return cluster. More simply, we can reduce the global std of the return by subtracting a baseline from its expectation, a common technique in traditional RL. Concretely, we replace the raw return with the advantage estimated by IQL (Kostrikov et al., 2021) and the updated consistency is presented in Table 2. And the corresponding performance improvement is exhibited in Table 4.

Table 2: The return consistency with advantage from IQL rather then original return.

| *Low* Datasets | HP-mr | AT-md | MZ-l |
|---|---|---|---|
| Return Range $\rightarrow$ Advantage Range | $[-1.25, 3.00] \rightarrow [-2.86, 1.25]$ | $[-0.21, 4.73] \rightarrow [-1.69, 1.96]$ | $[-0.32, 5.67] \rightarrow [-2.23, 6.05]$ |
| Return Mean $\rightarrow$ Advantage Mean | $2.05 \rightarrow 0.87$ | $0.06 \rightarrow 1.70$ | $3.16 \rightarrow 4.26$ |
| Return Std $\rightarrow$ Advantage Std | $0.94 \rightarrow \mathbf{0.11}$ | $1.12 \rightarrow \mathbf{0.10}$ | $1.00 \rightarrow \mathbf{0.01}$ |
| *consistency* | *Low* $\rightarrow$ **High** | *Low* $\rightarrow$ **High** | *Low* $\rightarrow$ **High** |

## 4 RELATED WORK

Offline Reinforcement Learning (Levine et al., 2020) breaks free from the traditional paradigm of online interaction (Sutton et al., 1998) and learns policy from fixed offline dataset collected by arbitrary or even unknown process (Lange et al., 2012; Fu et al., 2020). Most offline RL algorithms are developed based on classical online algorithms, such as CQL (Kumar et al., 2020) based on SAC (Haarnoja et al., 2018), TD3+BC (Fujimoto & Gu, 2021) based on TD3 (Fujimoto et al., 2018) and BPPO (Zhuang et al., 2023) based on PPO (Schulman et al., 2017). In contrast, Decision Transformer (DT) (Chen et al., 2021) directly maximizes the action likelihood, solving offline RL from supervised sequence modeling paradigm. Following upside-down RL (Srivastava et al., 2019; Schmidhuber, 2019), DT considers returns when predicting the action. Some works equip DT with classical RL components including dynamics programming (Yamagata et al., 2023), critic guidance (Wang et al., 2024; Hu et al., 2024), return maximization (Zhuang et al., 2024), online finetuning (Zheng et al., 2022) and trajectory stitching (Wu et al., 2023). On the other hand, DT is investigated from supervised learning perspective such as unsupervised pretraining (Xie et al., 2023; Carroll et al., 2022) and scaling ability (Lee et al., 2022; Shridhar et al., 2023). As for model architecture, LSTM (Siebenborn et al., 2022), one-dimension convolution network (Kim et al., 2023; Yan et al., 2024) and linear RNN (David et al., 2022; Cao et al., 2024; Ota, 2024; Lv et al., 2024; Huang et al., 2024) are adopted to replace the transformer (Vaswani et al., 2017) in DT. Some work also bring the critic or advantage

back to sequence modeling, such as Q-learning DT (Yamagata et al., 2023), CGDT (Wang et al., 2024), ACT (Gao et al., 2024). Besides, QT (Hu et al., 2024) using the Q-function as the loss of sequence modeling, which strictly speaking belongs to traditional RL with transformer policy.

# 5 RESULTS AND DISCUSSION

In this section, we present the performance of max-return sequence modeling across 9 datasets, 3 architectures, and 4 different context lengths. Based on this, we conduct an in-depth analysis to uncover the underlying principles of performance, gradually validating the above Conjectures 3. This leads us to draw conclusions regarding the relationship between performance, context length, model architecture, and return consistency. Finally, we demonstrate the significant improvement in trajectory stitching performance of Rein*for*mer after the return consistency is improved.

## 5.1 MAIN RESULTS

Table 3: The normalized score of max-return sequence modeling on 9 datasets (HC-me, HP-mr, WK-m, AT-mp, AT-md, KC-p, MZ-l, P-h, P-c) with 3 different architectures and 4 context-lengths ($K$). Datasets with **high** return consistency are red while *Low* is green . We report the mean of normalized score for five seeds. For each seed, the normalized score is calculated by the mean of 10 evaluation trajectories for Gym and Adroit while 100 for Antmaze, Maze2d and Kitchen. The scores below IQL are in gray. The best result is blue and the **bold** result means the best result among one sequence model with different $K$. The last row represents how many results outperforms IQL.

| model | $K$ | HC-me | HP-mr | WK-m | AT-mp | AT-md | KC-p | MZ-l | P-h | P-c |
|---|---|---|---|---|---|---|---|---|---|---|
| Rein*for*mer | 2 | 91.23 | **70.92** | 79.84 | **5.80** | 2.00 | 68.05 | 61.80 | 62.77 | 64.49 |
| Rein*for*mer | 5 | 90.99 | 68.80 | **79.91** | 4.20 | 3.40 | 73.00 | 64.95 | **75.15** | **86.55** |
| Rein*for*mer | 10 | 91.87 | 53.02 | 79.82 | 3.80 | **5.60** | **74.05** | 62.00 | 68.25 | 75.17 |
| Rein*for*mer | 20 | **92.81** | 40.84 | 72.25 | 1.60 | 4.20 | 66.20 | **64.99** | 71.94 | 74.79 |
| Reinconver | 2 | 91.83 | 84.24 | 72.28 | 6.20 | **5.40** | 65.20 | 32.69 | 73.64 | 68.52 |
| Reinconver | 5 | 92.26 | **84.44** | 74.09 | **7.80** | 4.20 | 34.55 | 22.45 | **82.23** | 71.68 |
| Reinconver | 10 | **92.90** | 54.02 | **75.88** | 4.40 | 5.20 | **65.85** | 34.74 | 76.27 | 62.58 |
| Reinconver | 20 | 92.78 | 49.22 | 75.38 | 2.00 | 2.60 | 65.25 | 32.39 | 75.29 | **83.38** |
| Reimba | 2 | 91.79 | **81.95** | 77.81 | 5.20 | 2.60 | 40.75 | **59.00** | 84.89 | 59.60 |
| Reimba | 5 | 92.91 | 74.24 | **80.03** | 12.40 | 5.00 | **45.10** | 41.04 | 82.91 | **71.28** |
| Reimba | 10 | **93.05** | 55.99 | 75.59 | 13.80 | 5.00 | 29.70 | 43.59 | **97.31** | 71.02 |
| Reimba | 20 | 92.42 | 49.47 | 73.35 | **15.60** | **9.00** | 29.05 | 43.14 | 91.61 | 70.57 |
| IQL | 1 | 86.70 | **94.70** | 78.30 | **65.80** | **73.80** | 46.30 | 61.70 | 71.50 | 37.30 |
| | | (12/12) | (0/12) | (4/12) | (0/12) | (0/12) | (6/12) | (4/12) | (10/12) | (12/12) |

Table 3 presents the performance of max-return sequence modeling with different parameters on diverse data distribution. We also compare the performance of sequence modeling algorithms with the most classic offline RL algorithm IQL (Kostrikov et al., 2021), highlighting scores below IQL in gray. IQL significantly outperforms sequence modeling on datasets with *Low* return consistency including HP-mr, AT-mp, AT-md and MZ-l. Moreover, these datasets are heavily emphasize trajectory stitching ability, which corresponds to the definition of *Low* consistency. In contrast, datasets with **high** consistency either matches or even surpasses the performance of IQL. This experiment results confirm the validity of our first conjecture.

> **Conclusion I**
>
> Return consistency significantly impacts the performance of max-return sequence modeling. *Low* return consistency datasets often exhibit notably inferior performance while **high** consistency datasets achieve comparable performance and even surpass RL.

## 5.2 CONTEXT LENGTH

In this section, we investigate the impact of the historical sequence length, also known as context length $K$, on performance. Sequence modeling and MDPs have distinct perspectives on the historical trajectory when predicting actions, making context length a crucial factor in sequence modeling.

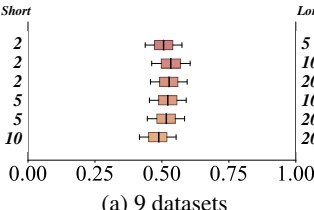 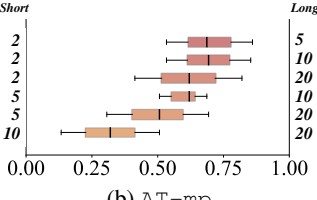 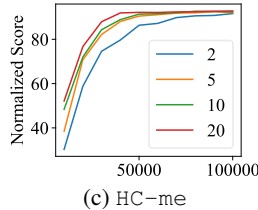

(a) 9 datasets        (b) `AT-mp`        (c) `HC-me`

Figure 4: (a) represents the probability of the left $K$ superior to the right one across all datasets. (b) represents the probability on `AT-mp` (c) represents evaluation scores with different $K$ on `HC-me`.

### 5.2.1 CONTEXT LENGTH COMPARISON

We plot the performance curves and the return distribution on `HP-mr` in Figure 3. The quality of `HP-mr` is widely distributed, ranging from random to expert, with a peak less than 20 normalized score. The distribution of `HP-mr` is akin to an online replay buffer, which places higher demands on learning from suboptimal trajectories. Correspondingly, a smaller context length aligns more closely with the Markov Decision Process (MDP) framework, and thus performs better.

Upon considering all datasets, it becomes evident that no context length is universally applicable across 9 datasets (Figure 4a). For high-quality datasets, such as `HC-me`, a longer context length facilitates better training convergence and leads to improved score in Figure 4c. For trajectory stitching, shorter trajectories are preferred 4b. Taking into

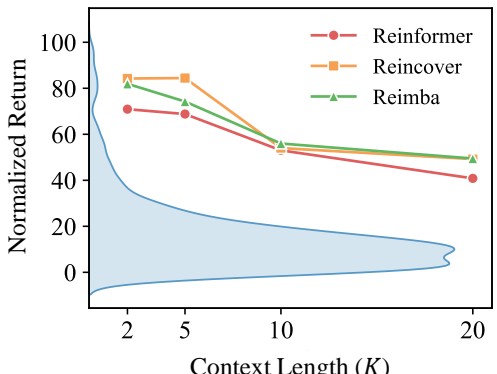

Figure 3: The data distribution (blue shade) and normalized evaluation score on `HP-mr`.

account longer historical trajectories increases the influence of past actions on subsequent behaviors, which may hinder the adoption of trajectories that deviate from historical ones. This is detrimental to the stitching process. In other words, longer historical trajectories can be seen as conservatism.

Based on the experiments outlined above, we can derive the second following conclusion:

> **Conclusion II**
>
> For datasets with **high** consistency, longer context length $K$ generally lead to better performance. Conversely, shorter $K$ tend to be more effective for *Low* consistency.

### 5.2.2 LONG TRAINING CONTEXT LENGTH WHILE SHORT INFERENCE CONTEXT LENGTH

All previous models have considered the historical trajectory length to be the same during inference as in the training phase. ODT (Zheng et al., 2022) discovers that, in some cases, a shorter sequence length during inference can help improve performance. EDT (Wu et al., 2023) also proposes the concept of dynamically adjusting the sequence length during inference. Inspired by these, we explore the performance of masking some historical tokens with the model trained by context length $K = 20$.

During the inference phase, we use historical trajectories of length $K_1 < 20$, padding the empty tokens with zeros to meet the model's input length requirement. This can be viewed as masking a segment of length $20 - K_1$. In Figure 5, as the horizontal axis increases, the input trajectory length $K_1$ decreases, while the zero-padded portion increases accordingly. Notably, the model's performance improves significantly under these conditions, surpassing all the different $K$.

### 5.3 ARCHITECTURE

In this section, we explore the impact of model architecture on the performance of sequence modeling. Prior research has indicated that the sequence modeling with Convformer (Kim et al., 2023) and Mamba architecture (Ota, 2024; Cao et al., 2024; Lv et al., 2024; Huang et al., 2024) outperform transformer. However, these conclusions were drawn without considering the diverse data distribution. Therefore, we will re-examine these findings across 9 datasets.

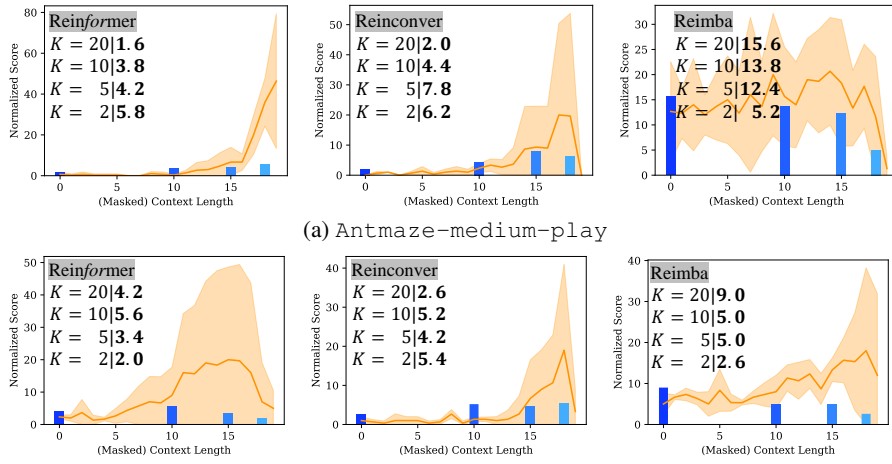

(a) `Antmaze-medium-play`

(b) `Antmaze-medium-diverse`

Figure 5: This figure displays the performance of masking the first $(20 - K_1)$ tokens in a sequence model with $K = 20$. The averages and corresponding standard deviations of three seeds are represented by the solid yellow line and its shaded area. Additionally, we compare this with models trained and evaluated normally with $K = 20, 10, 5, 2$ (blue bar values). The horizontal axis increases from left to right as the masked tokens increases and the remaining context length $K_1$ decreases.

### 5.3.1 ATTENTION ON HISTORICAL TOKENS

We analyze which part of the historical trajectory different model constructions specifically focus on. We selected the model trained with $K = 10$ on the Antmaze environment. Let $t$ represent the time step of the current token, and $t - 9$ represents the token furthest from the current time step. By masking a token at a certain position with 0, we calculate the difference between the masked output and the original output. This difference can, to some extent, reflect the importance of the masked token to the current output value. Then, based on this difference, we can determine whether the

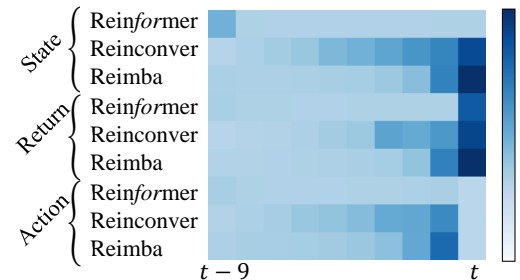

Figure 6: This heatmap illustrates the impact of token zero masking on the final output.

model pays more attention to global or local information. We have plotted the heatmaps of the differences in state, return, and action for the three models in the right figure.

The heatmap reveals that Reinconver and Reimba exhibit a significant increase in values at the current timestep, indicating the focus on local information. In contrast, the Rein*for*mer does not show a marked rise in differences, suggesting that the impact of masking any token is relatively uniform. Thus, Rein*for*mer focuses on global information.

### 5.3.2 ARCHITECTURE COMPARISON

In Figure 7, we illustrate the probability that the architecture on the left outperforms the right one. The closer the box is to the right side, the better the performance of the model on the left, and vice versa. A central position indicates that the two architectures have comparable performance. Considering the nine datasets collectively, no single model demonstrates an absolute advantage. In other words, the superiority of a model cannot be discussed independently of the characteristics of the dataset.

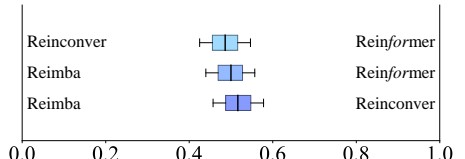

Figure 7: The improvement probability of the architectures across all the 9 datasets.

On the `MZ-l` dataset, the Rein*for*mer demonstrates a significant advantage. This is because the maze-large dataset inherently exhibits non-Markovian properties, where decisions based on the current state are correlated with historical waypoints. The Reinformer's focus on global historical trajectory information is particularly adept at considering and utilizing waypoint-related infor-

mation effectively. In contrast, on the Antmaze-medium-play dataset, which emphasizes trajectory stitching, models like Reincover and Reimba that focus on local information perform better. This is attributed to the fact that extensive historical sequence information leads to more conservative model outputs, reducing the likelihood of generating new decisions that deviate from historical trajectories.

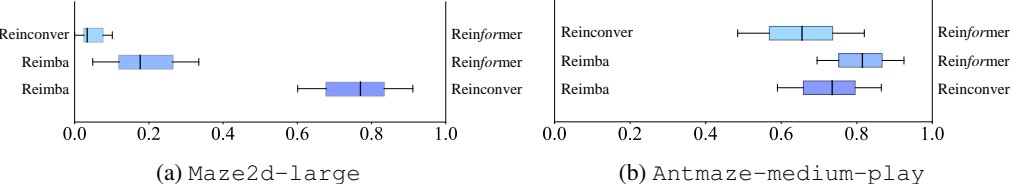

(a) `Maze2d-large`                    (b) `Antmaze-medium-play`

Figure 8: The probability of the model on the left superior to the model on the right across (a) `Maze2d-large` and (b) `Antmaze-medium-play`.

> **Conclusion III**
>
> For **high** consistency datasets, architectures that focus on global information tend to better. Conversely, it is preferable to focus on local for *low* consistency datasets.

## 5.4 RESULTS WITH ADVANTAGE

As indicated in Table 2, replacing the return with the advantage can significantly reduce the std of the cluster with the highest mean, thereby transforming the scenario from *low* return consistency to **high**. In Section 5.1, we have already revealed the positive correlation between the performance of max-return sequence modeling and return consistency. Therefore, we investigate here whether the performance of max-return sequence modeling, with the aid of the advantage, far surpasses that based on the return. This can also serve as a validation of Conclusion I.

**Baselines**   Rein*for*mer-best denotes the best performance achieved for each dataset in Table 1, regardless of the context length and model architecture. Rein*for*mer + $A$ indicates that the returns-to-go are replaced with the advantage during training. Elastic Decision Transformer (EDT) (Wu et al., 2023) explicitly consider trajectory stitching through dynamically adjusts historical trajectories during inference. ACT (Gao et al., 2024) also adopts the advantage of IQL while Q-learning DT (Yamagata et al., 2023) adopts the conservative Q from CQL (Kumar et al., 2020).

Table 4: Performance on *low* return consistency datasets with other baselines. The best results among sequence models are **bold**. Some results are reproduced using public code.

| Datasets | IQL | Rein*for*mer-best | Rein*for*mer + $A$ | DT | ODT | EDT | ACT | Q-learning DT |
|---|---|---|---|---|---|---|---|---|
| HP-mr | 94.7 | 84.44 | $88.24 \pm 5.21 (+ 3.80)$ | 82.7 | 86.6 | 91.2 | **98.4** | 52.1 |
| AT-mp | 65.8 | 15.60 | $\mathbf{43.60 \pm 13.89} (+28.00)$ | 0.8 | 0.0 | 0.0 | 1.8 | — |
| AT-md | 73.8 | 9.00 | $\mathbf{56.50 \pm 6.36} (+47.50)$ | 0.5 | 0.0 | 0.0 | 2.4 | — |
| MZ-l | 61.7 | 64.99 | $\mathbf{116.92 \pm 48.30} (+51.93)$ | 35.7 | 39.4 | 26.8 | 58.3 | 31.0 |

As anticipated before, replacing the return with the advantage significantly enhances return consistency, which in turn leads to a substantial improvement in trajectory stitching capabilities. Thus, we have not only elucidated the reasons behind the suboptimal performance of max-return sequence models but also proposed a simple yet highly effective solution.

## 6 CONCLUSION

In summary, our investigation into max-return sequence modeling reveals that return consistency is a crucial factor influencing performance. Datasets with **high** return consistency tend to achieve better performance in max-return sequence modeling. These datasets also benefit more from longer context lengths and architectures that emphasize global information. Additionally, we find that replacing the return with the advantage function can enhance return consistency, leading to significant performance improvements and stronger stitching ability. These findings provide valuable insights for advancing sequence modeling and bridging the gap with classical offline RL algorithms.

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

## A   LLMs Usage Statement

We used large language models (LLMs) solely for the purpose of grammar checking, sentence polishing, and improving the overall readability of the manuscript. The use of LLMs was strictly limited to linguistic refinement and did not involve any aspect of the core research methodology. All technical contributions, ideas, analyses, and conclusions presented in this paper are entirely the work of the authors.

## B   Limitations

Our work, while advancing max-return sequence modeling, has limitations. The introduction of the advantage function, though beneficial, adds computational overhead, potentially limiting scalability. Additionally, our clustering-based metric for return consistency is simplistic and may not fully capture data complexity.

## C   Diverse Datasets

We selected nine representative datasets from the widely-used offline benchmark D4RL to evaluate the sequence modeling, which are detailed as follows:

- `Halfcheetah-medium-expert`, `Hopper-medium-replay` and `Walker2d-medium`: The abbreviations are respectively `HC-me`, `HP-mr` and `WK-m`. For `Gym` tasks, we select only one dataset from each environment, which encompasses three distinct data distributions. The "medium-replay" dataset consists of samples in the replay buffer observed during online training until the policy reaches the "medium" level, approximately 1/3 the performance of the "expert".

- `Antmaze-medium-play` and `Antmaze-medium-diverse`: The abbreviations are respectively `AT-mp` and `AT-md`. `Antmaze` datasets have a sparse reward to show if the ant reach the goal in the maze. The `medium` dataset requires the algorithm to navigate to the target point by stitching the suboptimal trajectories into the successful trajectories. These datasets require the trajectory stitching ability, which is particularly challenging for sequence modeling.

- `Kitchen-partial`: The abbreviation is `KC-p`. The desired goals are to complete 4 subtasks: open the microwave, move the kettle, flip the light switch, and slide open the cabinet door. The "partial" dataset includes subtrajectories where the 4 target subtasks are completed in sequence.

- `maze2d-large`: The abbreviation is `MZ-l`. The dataset is collected by a PD controller that memorizes the reached waypoints during data collection, so the Markov property does not hold.

- `Pen-human` and `Pen-cloned`: The abbreviations are respectively `P-h` and `P-c`. This environment controls a 24-DoF simulated Shadow Hand robot to twirl a pen. `Human` dataset contains 25 trajectories of expert demonstration. `Cloned` dataset uses a 50-50 split between demonstration data and trajectories sampled from a behavior cloned policy trained on the demonstrations.

In summary, these 9 datasets each have their own distinctive features. In addition to the three commonly used `Gym` datasets, our selection also encompasses `Antmaze` datasets that emphasize trajectory stitching, `Kitchen` dataset that includes partial expert demonstration segments, `maze` dataset highlighting non-Markovian properties, and `Pen` dataset that incorporates expert demonstrations.

## D   Architectures

The implementation of policy $\pi$ is based on the sequence model and the predictions $\hat{g}_t$, $\hat{a}_t$ are achieved through an autoregressive approach. Moving forward, we primarily consider three architectural

variants: the Transformer (Vaswani et al., 2017), One-dimensional convolution layers (Conv) (Yu et al., 2022), and the linear Recurrent Neural Network Mamba (Gu & Dao, 2023).

- Rein*for*mer is based on the Transformer architecture, built upon the self-attention mechanism, equipped with multiple attention heads and stacked encoder-decoder structures, can adeptly captures long-range dependencies. The Decoder module within the Transformer has found wide application in NLP and Offline Reinforcement Learning tasks, as demonstrated by models like Decision Transformer. The equation presented exemplifies the attention mechanism used in the Transformer framework:

$$Attention(Q, K, V) = softmax\left(\frac{QK^T}{\sqrt{d_k}}\right)V \tag{7}$$

- Reinconver is based on 1D CNN. In the field of sequence modeling, 1D convolutions play a role in extracting local patterns and features from sequences, aiding in learning positional invariance. It is worth mentioning that, positional information is inherently included during the convolution process due to the local receptive field property, so we did not add positional embedding to Reinconver.

- Reimba is based on the linear RNN Mamba. Inspired by continuous-time systems, Mamba models sequences or one-dimensional functions through a recurrent mapping process. Like S4, Mamba uses a hidden state representation, where the hidden state evolves through time as the system processes inputs. These equations describe the time evolution of the hidden state, with:

$$h'(t) = Ah(t) + Bx(t), \quad y(t) = Ch(t), \tag{8}$$

where $A \in \mathbb{R}^{N \times N}$ is the evolution matrix, $B \in \mathbb{R}^{N \times 1}$ and $C \in \mathbb{R}^{1 \times N}$ are projection matrices that govern how inputs and hidden states are transformed into outputs. In the discrete case, Mamba uses techniques similar to S4, where continuous parameters $A$ and $B$ are discretized, enabling the model to handle sequences. This leads to a discrete-time variant of the ODEs:

$$h_t = \bar{A}h_{t-1} + \bar{B}x_t, \quad y_t = Ch_t, \tag{9}$$

where $\bar{A} = \exp(\Delta)A$ and $\bar{B} = (\Delta A)^{-1}(\exp(\Delta A) - I)(\Delta B)$, with $\Delta$ representing a timescale parameter. Mamba introduces a selective scan mechanism, allowing it to dynamically evolve hidden states based on input data, which ensures Mamba efficiently captures long-range dependencies while maintaining computational efficiency for long sequences. Mamba is currently a hot contender in the fields of CV and NLP. At the same time, since Mamba is essentially a type of RNN-like structure capable of extracting positional information, we did not include positional embedding to Reimba.

## E    INFLUENCE OF POSITIONAL EMBEDDING

As previously mentioned, we do not use positional embedding in Reinconver and Reimba. We believe the positional embedding is harmful to trajectory stitching. Positional embedding are directly added to embedded state, returns and action tokens. As a result, the same input sequences become different at different timesteps, which is harmful to stitching under similar state sequences. This is supported by "w/o" results in Table 5 especially on short Context Length.

Positional embedding facilitates effective information extraction from long sequences. On `Hp-mr` dataset, the advantage of long sequences with positional embedding in information extraction outweighs their disadvantage in trajectory stitching, causing performance improvement with large $K$. But on `AT-mp` that heavily emphasizes stitching, the advantage in information extraction does not surpass the disadvantage in trajectory stitching, even in the scenario of large $K$.

## F    HYPERPARAMETERS

Hyperparameters used during model training are as follows:

| model | $K$ | HP-mr | | | AT-mp | | |
|---|---|---|---|---|---|---|---|
| | | w/o | w/ | $\Delta$ | w/o | w/ | $\Delta$ |
| Reinconver | 2 | 84.24 | 67.55 | -19.81% | 6.20 | 2.67 | -56.94% |
| Reinconver | 5 | 84.44 | 72.74 | -13.86% | 7.80 | 7.00 | -10.26% |
| Reinconver | 10 | 54.02 | 75.52 | +39.80% | 4.40 | 2.33 | -47.05% |
| Reinconver | 20 | 49.22 | 68.87 | +39.92% | 2.00 | 1.00 | -50.00% |
| Reimba | 2 | 81.95 | 76.87 | -6.20% | 5.20 | 8.33 | +60.19% |
| Reimba | 5 | 74.24 | 77.23 | +4.03% | 12.40 | 11.00 | -11.29% |
| Reimba | 10 | 55.99 | 60.94 | +8.84% | 13.80 | 10.00 | -27.54% |
| Reimba | 20 | 49.47 | 70.03 | +41.56% | 15.60 | 11.00 | -29.49% |

Table 5: The normalized scores of Reinconver and Reimba without and with positional embedding. Default Reimba and Reinconver did not include positional embedding.

| env name | model | $K$ | tau | train step | learning rate | normalized score |
|---|---|---|---|---|---|---|
| HC-me | Rein*for*mer | 2 | 0.99 | 10w | 0.0001 | 91.23 |
| | Rein*for*mer | 5 | 0.99 | 10w | 0.0001 | 90.99 |
| | Rein*for*mer | 10 | 0.99 | 10w | 0.0001 | 91.87 |
| | Rein*for*mer | 20 | 0.99 | 10w | 0.0001 | 92.81 |
| | Reinconver | 2 | 0.99 | 10w | 0.0001 | 91.83 |
| | Reinconver | 5 | 0.99 | 10w | 0.0001 | 92.26 |
| | Reinconver | 10 | 0.99 | 10w | 0.0001 | 92.9 |
| | Reinconver | 20 | 0.99 | 10w | 0.0001 | 92.78 |
| | Reimba | 2 | 0.99 | 10w | 0.0001 | 91.79 |
| | Reimba | 5 | 0.99 | 10w | 0.0001 | 92.91 |
| | Reimba | 10 | 0.99 | 8w | 0.0001 | 93.05 |
| | Reimba | 20 | 0.99 | 4w | 0.0001 | 92.42 |
| HP-mr | Rein*for*mer | 2 | 0.999 | 9w | 0.0004 | 70.92 |
| | Rein*for*mer | 5 | 0.999 | 9w | 0.0004 | 68.80 |
| | Rein*for*mer | 10 | 0.999 | 9w | 0.0004 | 53.02 |
| | Rein*for*mer | 20 | 0.999 | 9w | 0.0004 | 40.84 |
| | Reinconver | 2 | 0.999 | 8w | 0.0004 | 84.24 |
| | Reinconver | 5 | 0.999 | 8w | 0.0004 | 84.44 |
| | Reinconver | 10 | 0.999 | 8w | 0.0004 | 54.02 |
| | Reinconver | 20 | 0.999 | 8w | 0.0004 | 49.22 |
| | Reimba | 2 | 0.999 | 5w | 0.0004 | 81.95 |
| | Reimba | 5 | 0.999 | 10w | 0.0004 | 74.24 |
| | Reimba | 10 | 0.999 | 10w | 0.0004 | 55.99 |
| | Reimba | 20 | 0.999 | 10w | 0.0004 | 49.47 |
| WK-m | Rein*for*mer | 2 | 0.99 | 2w | 0.0001 | 79.84 |
| | Rein*for*mer | 5 | 0.99 | 1.5w | 0.0001 | 79.91 |
| | Rein*for*mer | 10 | 0.99 | 1.5w | 0.0001 | 79.82 |
| | Rein*for*mer | 20 | 0.99 | 2w | 0.0001 | 72.25 |
| | Reinconver | 2 | 0.99 | 7w | 0.0001 | 72.28 |
| | Reinconver | 5 | 0.99 | 7w | 0.0001 | 74.09 |
| | Reinconver | 10 | 0.99 | 7w | 0.0001 | 75.88 |
| | Reinconver | 20 | 0.99 | 7w | 0.0001 | 75.38 |
| | Reimba | 2 | 0.999 | 1w | 0.0001 | 77.81 |
| | Reimba | 5 | 0.999 | 1.5w | 0.0001 | 80.03 |
| | Reimba | 10 | 0.999 | 1w | 0.0001 | 75.59 |
| | Reimba | 20 | 0.999 | 1w | 0.0001 | 73.35 |

| env name | model | $K$ | tau | train step | learning rate | normalized score |
|---|---|---|---|---|---|---|
| KC-p | Rein*for*mer | 2 | 0.9 | 20w | 0.0001 | 68.05 |
| | Rein*for*mer | 5 | 0.9 | 20w | 0.0001 | 73 |
| | Rein*for*mer | 10 | 0.9 | 20w | 0.0001 | 74.05 |
| | Rein*for*mer | 20 | 0.9 | 10w | 0.0001 | 66.2 |
| | Reinconver | 2 | 0.99 | 20w | 0.0001 | 65.2 |
| | Reinconver | 5 | 0.99 | 20w | 0.0001 | 34.55 |
| | Reinconver | 10 | 0.99 | 20w | 0.0001 | 65.85 |
| | Reinconver | 20 | 0.99 | 20w | 0.0001 | 65.25 |
| | Reimba | 2 | 0.99 | 6w | 0.0001 | 40.75 |
| | Reimba | 5 | 0.99 | 5w | 0.0001 | 45.1 |
| | Reimba | 10 | 0.99 | 4w | 0.0001 | 29.7 |
| | Reimba | 20 | 0.99 | 2w | 0.0001 | 29.05 |
| MZ-l | Rein*for*mer | 2 | 0.999 | nan | 0.0004 | nan |
| | Rein*for*mer | 5 | 0.999 | 10w | 0.0004 | 64.95 |
| | Rein*for*mer | 10 | 0.999 | 10w | 0.0004 | 62 |
| | Rein*for*mer | 20 | 0.999 | 10w | 0.0004 | 64.99 |
| | Reinconver | 2 | 0.999 | 10w | 0.0004 | 32.69 |
| | Reinconver | 5 | 0.999 | 10w | 0.0004 | 22.45 |
| | Reinconver | 10 | 0.999 | 10w | 0.0004 | 34.74 |
| | Reinconver | 20 | 0.999 | 10w | 0.0004 | 32.39 |
| | Reimba | 2 | 0.999 | 10w | 0.0004 | 59.00 |
| | Reimba | 5 | 0.999 | 5w | 0.0004 | 41.04 |
| | Reimba | 10 | 0.999 | 5w | 0.0004 | 43.59 |
| | Reimba | 20 | 0.999 | 5w | 0.0004 | 43.14 |
| P-h | Rein*for*mer | 2 | 0.9 | 4w | 0.0001 | 62.77 |
| | Rein*for*mer | 5 | 0.9 | 4w | 0.0001 | 75.15 |
| | Rein*for*mer | 10 | 0.9 | 10w | 0.0001 | 68.25 |
| | Rein*for*mer | 20 | 0.9 | 10w | 0.0001 | 71.94 |
| | Reinconver | 2 | 0.99 | 4w | 0.0001 | 73.64 |
| | Reinconver | 5 | 0.99 | 4w | 0.0001 | 82.23 |
| | Reinconver | 10 | 0.99 | 5w | 0.0001 | 76.27 |
| | Reinconver | 20 | 0.99 | 5w | 0.0001 | 75.29 |
| | Reimba | 2 | 0.99 | 4w | 0.0001 | 84.89 |
| | Reimba | 5 | 0.99 | 4w | 0.0001 | 82.91 |
| | Reimba | 10 | 0.99 | 4w | 0.0001 | 97.31 |
| | Reimba | 20 | 0.99 | 4w | 0.0001 | 91.61 |
| P-c | Rein*for*mer | 2 | 0.9 | 5w | 0.0001 | 64.49 |
| | Rein*for*mer | 5 | 0.9 | 5w | 0.0001 | 86.55 |
| | Rein*for*mer | 10 | 0.9 | 5w | 0.0001 | 75.17 |
| | Rein*for*mer | 20 | 0.9 | 5w | 0.0001 | 74.79 |
| | Reinconver | 2 | 0.99 | 5w | 0.0001 | 68.52 |
| | Reinconver | 5 | 0.99 | 5w | 0.0001 | 71.68 |
| | Reinconver | 10 | 0.99 | 5w | 0.0001 | 62.58 |
| | Reinconver | 20 | 0.99 | 5w | 0.0001 | 83.38 |
| | Reimba | 2 | 0.99 | 5w | 0.0001 | 59.60 |
| | Reimba | 5 | 0.99 | 5w | 0.0001 | 71.28 |
| | Reimba | 10 | 0.99 | 5w | 0.0001 | 71.02 |
| | Reimba | 20 | 0.99 | 5w | 0.0001 | 70.57 |

| env name | model | $K$ | tau | learning rate | normalized score |
|---|---|---|---|---|---|
| AT-mp | Rein*for*mer | 2 | 0.999 | 0.0008 | 5.8 |
| | Rein*for*mer | 5 | 0.999 | 0.0008 | 4.2 |
| | Rein*for*mer | 10 | 0.999 | 0.0008 | 3.8 |
| | Rein*for*mer | 20 | 0.999 | 0.0008 | 1.6 |
| | Reinconver | 2 | 0.999 | 0.0008 | 6.2 |
| | Reinconver | 5 | 0.999 | 0.0008 | 7.8 |
| | Reinconver | 10 | 0.999 | 0.0008 | 4.4 |
| | Reinconver | 20 | 0.999 | 0.0008 | 2 |
| | Reimba | 2 | 0.999 | 0.0008 | 5.2 |
| | Reimba | 5 | 0.999 | 0.0008 | 12.4 |
| | Reimba | 10 | 0.999 | 0.0008 | 13.8 |
| | Reimba | 20 | 0.999 | 0.0008 | 15.6 |
| AT-md | Rein*for*mer | 2 | 0.999 | 0.0008 | 2 |
| | Rein*for*mer | 5 | 0.999 | 0.0008 | 3.4 |
| | Rein*for*mer | 10 | 0.999 | 0.0008 | 5.6 |
| | Rein*for*mer | 20 | 0.999 | 0.0008 | 4.2 |
| | Reinconver | 2 | 0.999 | 0.0008 | 5.4 |
| | Reinconver | 5 | 0.999 | 0.0008 | 4.2 |
| | Reinconver | 10 | 0.999 | 0.0008 | 5.2 |
| | Reinconver | 20 | 0.999 | 0.0008 | 2.6 |
| | Reimba | 2 | 0.999 | 0.0008 | 2.6 |
| | Reimba | 5 | 0.999 | 0.0008 | 5 |
| | Reimba | 10 | 0.999 | 0.0008 | 5 |
| | Reimba | 20 | 0.999 | 0.0008 | 9 |

Table 6: The normalized scores of Reinconver and Reimba with and without positional embeddings. Original Reimba and Reinconver did not include positional embedding. The $\Delta$ represents the change in score when positional embedding is added.

| | | | WK-m | | | HC-me | | |
|---|---|---|---|---|---|---|---|---|
| model | $K$ | no_pos | pos | $\Delta$ | no_pos | pos | $\Delta$ |
| Reinconver | 5 | 74.09 | 75.48 | +1.88% | 92.26 | 91.80 | -0.50% |
| Reimba | 5 | 80.03 | 74.73 | -6.62% | 92.91 | 91.57 | -1.44% |

Table 7: window size for Reinconver

| $K$ | 2 | 5 | 10 | 20 |
|---|---|---|---|---|
| window size | 4 | 6 | 10 | 20 |

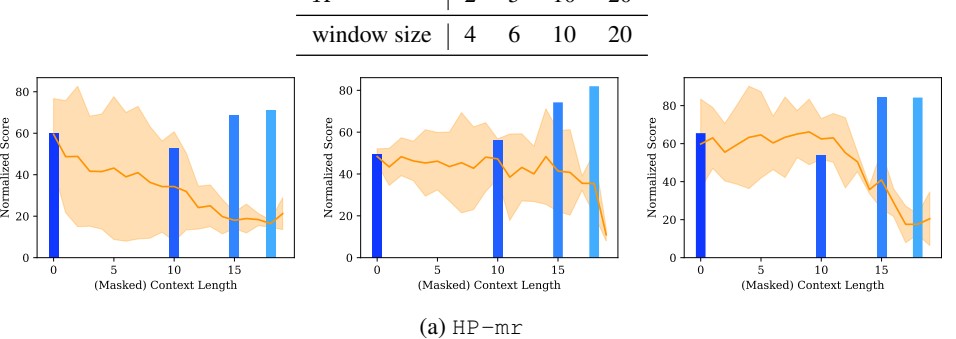

(a) HP-mr

Figure 9: This figure displays the performance of masking the first $(20 - K_1)$ tokens in a sequence model with $K = 20$. We show the averages and corresponding standard deviations of three seeds evaluated in the *HP-mr* environment 10 times (represented by the solid yellow line and its shaded area). Additionally, we compare this with models trained and evaluated normally with a length of $20, 10, 5, 2$ (blue bar values). The horizontal axis increases from left to right as the number of masked tokens increases and the remaining context length $K_1$ decreases.

