# OpenReview forum: "Unraveling Max-Return Sequence Modeling via Return Consistency"
_ICLR.cc/2026/Conference — ICLR 2026 Conference Withdrawn Submission_

### Official Review · Reviewer_grBi · 2025-10-28

**Soundness:** 2
**Presentation:** 2
**Contribution:** 3
**Rating:** 6
**Confidence:** 3

**Summary:**

This paper tackles the performance/stability problem of training Decision Transformers in the offline RL setting. The authors argue that "return consistency" can explain certain failure modes and dependencies on the hyperparameters (e.g., context length/architecture).

**Strengths:**

The paper is overall easy to follow, and the idea of using return consistency as a predictor for performance is, to the best of my knowledge, a novel approach.
The hypotheses are supported by extensive experiments across various settings and environments.
The findings may be useful for practitioners to identify potential problems when training DTs.

**Weaknesses:**

1. There are details that are not fully explained in the paper, which makes some parts difficult to understand (see questions).
2. While the authors showed that return consistency can explain some behaviors of DTs, an explanation of why this is the case remains desirable.

Minor suggestions:

1. Line 160: I guess you don't really need an order between $R_t^{(1)}$ and $R_t^{(2)}$, but just that the two are not too close (e.g., a definition based on $|R_t^{(1)} - R_t^{(2)}|$.)
2. Line 169: $\mathfrak{G}$ is not defined.
3. There are various typos (e.g., line 60 "In summarize",  line 143 "concretely metric".)
4. AntMaze is sometimes referred to as AM or AT.

**Questions:**

1. In Fig1 (left), what are 1), 2) and 3)? These are not explained in the caption.
2. (Def 3.1) Since actions are chosen based on sequences, is it reasonable to consider sequences $(\tau, s_t, a_t, R_t)$ instead of just the current state?
3. For DBSCAN, how sensitive are the results towards the clustering parameters (e.g., eps)? Is it shared across environments?
4. I think the context length experiments (Sec 5.2) may be confounded by the degree to which the environments are partially observable (more partially observable environments can benefit more from longer context lengths). Have the authors considered this?
5. (Sec 5.3.1) It is not clear to me if masking a token with 0 is a good way to assess the importance of a token. Can the authors elaborate on this?

---

### Official Review · Reviewer_r5br · 2025-10-30

**Soundness:** 2
**Presentation:** 2
**Contribution:** 1
**Rating:** 2
**Confidence:** 4

**Summary:**

This paper investigates the supervised learning paradigm in offline reinforcement learning. Specifically, it examines the Reinformer’s max-return sequence modeling and introduces a metric called "return consistency"—based on the idea that nearby state–action pairs should yield similar returns. The study then correlates this metric with the performance of sequence-modeling policies, reports several empirical patterns, and analyzes the conditions under which high performance can be achieved. Finally, it proposes using advantage instead of the original return, to reduce variance, thereby further improving performance.

**Strengths:**

- This paper proposes the "return consistency" metric to analyze offline datasets and examine the relationships between different factors—including context length and model architectures. This work provides a new perspective for advancing offline reinforcement learning.
- The paper is well-motivated and easy to follow.

**Weaknesses:**

- The motivational example in Figure 1 considers a scenario where two data points with similar actions but different returns may cause the trained model to select the action leading to a low return. However, it omits a more common case: the model learns an average of the two actions, resulting in a wrong action that aligns with neither of the original actions. It would be better to experimentally verify which of these cases actually occurs.

- The proposed metric is straightforward and intuitively sound, but it lacks in-depth analysis. The results reveal certain phenomena and convey intuitive insights, yet the absence of explanations for the reasons behind these phenomena leaves readers confused about why they occur.

- Many parameters are missing from the experiments—for instance, the parameters for DBSCAN and IQL. Additionally, the experiments only use three or five seeds, and as shown in Figure 5, the results exhibit very high variance. This makes it unconvincing to draw definitive conclusions.

- Overall, this draft reads more like a technical report. It could be further improved by justifying the choice of methods (such as DBSCAN and IQL); reporting results across varying parameter settings alongside significance tests; and providing deeper analysis of the metric and the factors influencing its performance.

**Questions:**

- The first sentence of the abstract is confusing: "Offline reinforcement learning (RL) ..., enabling supervised solutions for offline RL." What specific meaning does this sentence intend to convey?
- States and actions carry distinct semantic meanings, which may impact different factors. For example, scenarios with similar states but slightly different actions, or similar actions but slightly different states, could lead to varying influences. Does the current approach—when clustering data using (s,a) pairs—account for these differences? Additionally, how was the DBSCAN method selected for clustering?
- Figure 2 uses Umap for dimensionality reduction, while DBSCAN is later employed for clustering. Is the clustering result generated by DBSCAN consistent with the pattern shown in Figure 2?
- A single (s,a) pair may correspond to different trajectories, ultimately leading to varying returns. This phenomenon is more likely to occur at early time steps than at later ones. Does this imply that the relationship between the proposed metric and final performance is tied to time steps, rather than being solely dependent on context length?
- Conclusion 2 claims: "For datasets with high consistency, longer context length K generally leads to better performance. Conversely, shorter K tends to be more effective for low consistency." Are there additional explanations for why this trend occurs?
- For further concerns regarding the work, please refer to the "Weaknesses" section.

---

### Official Review · Reviewer_gUhA · 2025-11-01

**Soundness:** 2
**Presentation:** 1
**Contribution:** 2
**Rating:** 4
**Confidence:** 4

**Summary:**

This paper investigates the performance bottleneck of max-return sequence modeling (Reinformer) in offline RL by introducing the concept of return consistency, which measures how well similar state-action pairs correspond to similar returns. In a problematic scenario during inference, when similar state-action pairs have very distinct RTGs, Reinformer outputs the higher RTG, but subsequently may generate the action paired with a lower RTG. The authors demonstrate that high return consistency leads to better performance on 9 D4RL datasets and propose that IQL-based advantage can further improve return consistency compared to the raw RTG. In addition, the authors claim that high return consistency is also compatible with longer context lengths and benefits from model architectures that capture global information.

**Strengths:**

- The notion of return consistency is intuitive and straightforward.
- This paper conducts a large batch of experiments by examining multiple model architectures and context length choices on multiple types of datasets.
- The figures and tables in this work are compact and informative.

**Weaknesses:**

1. __The overall presentation of this paper could be largely improved.__
    - In Section 2, **mathematical notations** are not rigorous and consistent. For example, $\tau_{t-K}$ includes transitions up to timestep $t+1$, but the DT loss in equation 1 predicts $a_t$ conditioned on $\tau_{t-K}$, which by your definition contains future information at timestep $t+1$. Also, in Line 96, conditioned on $\hat{R}_0$ and $s_0$, $\pi$ should predict for $a_0$ in consistence with your definition of the dataset $\mathcal{D}$.
    - I hardly followed **the presentation in Sections 5.2 and 5.3** when reading the paper. They feel digressive and are weakly connected to the central topic: return consistency. In my opinion, Section 5.2 conveys that a longer context may harm performance by not allowing the model to deviate from historical trajectories in the training dataset. Section 5.3 mainly shows that Reinformer focuses more on global sequence modeling, while Reinconver and Reimba focus on the local immediate modeling. How do these connect to return consistency and then lead to conclusions 2 and 3?
    - There exist contradictory arguments in the paper. For example, both MZ-l and AT-mp have low return-consistency and should favor an architecture that focuses on local information by conjecture 3. However, in Section. 5.3.2, the authors conclude that Reinformer works better for AT-mp, while Reincover and Reimba work better for MZ-l. This fails to lead to conclusion 3.
2. Table 3 seems to already show some evidence related to conjectures 2 and 3, **but the evidence is not enough.** In terms of conjecture 2, I do not observe a clear increasing performance when increasing context lengths on high return-consistency datasets, although increasing context lengths often leads to lower performance on those low return-consistency datasets. In terms of conjecture 3, I expect to see: Reinformer (which focuses on global information) works better on high return-consistency datasets, while Reinconver or Reimba (which focuses on local information) work better on low return-consistency datasets. However, the results on P-h and MZ-l do not match the expectation and thus may not support conjectures 2 and 3.
3. **It is not convincing enough that the return consistency metric should only focus on high-return regions.** High-return regions do not necessarily cover enough state and action spaces, so consistency for one region implies little for other regions. Could the authors justify their choice of the current metric for measuring return consistency? Are there any alternative definitions that might better reflect this concept? For example, would it make sense to average the return std of all clusters?
4. **What are the factors that influence the return consistency of a dataset?** In Sections 5.2 and 5.3, it seems that return consistency mingles with several additional concepts, including requirement of trajectory stitching ability, obedience to Markovian property, data suboptimality, partial observability, and environment stochasticity. Without disentangling these compounding factors, the analysis can be confusing and less evident.
5. **How does the concept of return consistency generalize to other sequence modeling algorithms?** This paper focuses on a particular algorithm, Reinformer, as it employs max-return predication. Will similar conclusions 1-3 hold for other related algorithms? For example, will original DT or some strong baseline like QT perform better on high return-consistency and worse on low consistency?

**Questions:**

Please refer to the weaknesses.

---

### Official Review · Reviewer_jye7 · 2025-11-03

**Soundness:** 3
**Presentation:** 3
**Contribution:** 3
**Rating:** 4
**Confidence:** 4

**Summary:**

The paper investigates why max‑return sequence modelling excels on some offline RL datasets and fails on others. It introduces return consistency and measures it via the std of returns in the highest‑return cluster after clustering state–action embeddings. Empirically,  high‑consistency datasets favour max‑return sequence modelling, longer context, and global‑attention architectures; low‑consistency datasets favour shorter effective context and local models. A simple variance‑reduction tweak substantially improves the metric and performance on previously low‑consistency tasks.

**Strengths:**

- Clear data-driven explanation of when max-return sequence modelling succeeds or fails, the “two regimes” are clearly visible
- Broad and systematic empirical investigation
- Actionable guidance has been provided and is informative
- Simple tweak (train on advantages) that measurably reduces the cluster spread and often lifts performance with very large gains on Maze2d-large
- Helpful architecture analysis: token-mask heat maps suggest Transformer behaves more “global,” Conv/Mamba more “local” aligning with the regime story.

**Weaknesses:**

- The return-consistency metric is pipeline-dependent and there’s no sensitivity analysis to these choices.
- Using dataset returns makes the diagnostic partly policy-dependent and entangles locality with trajectory stitching; interpretation could be tightened.
- The improvement relies on IQL advantages; no study of alternative critics/estimators or how advantage quality affects results; added compute isn’t profiled.
- There is no theoretical analysis that ties the measured dispersion to sequence-model error, findings are purely empirical; this is a significant weakness

**Questions:**

1. Clarify whether the return-consistency clustering is done in the raw state–action feature space or in the same embedding used for the UMAP visualisation, and report the DBSCAN hyperparameters (and any dimensionality-reduction choices). Can you include a brief sensitivity analysis, and consider a simple k-NN local-variance proxy as a cross-check?

2. Can you separate trajectory stitching effects from local smoothness by recomputing the consistency metric using a learned Q(s, a) or truncated returns around the decision step?

3. What is the compute and memory overhead of training with IQL-based advantages compared to the vanilla max-return setup?

4. Do advantages from other critics (e.g., CQL) or a fitted-value baseline yield similar gains? A small ablation should be able to show whether improvements are specific to IQL.

5. When training on advantages, what exactly is conditioned on at inference?

6. The token-masking heatmaps suggest different historical-focus patterns across architectures. Could you add a quantitative “attention localisation” metric across seeds to support this?

---

### Note · Authors · 2026-04-30

I have read and agree with the venue's withdrawal policy on behalf of myself and my co-authors.

---

### Meta-Review · Area_Chair_kZph · 2026-01-06

**Summary:**

This paper analyzes when max-return sequence modeling (e.g., Reinformer-style) succeeds or fails in offline RL, proposing a “return consistency” diagnostic (cluster-based return dispersion for similar state–action embeddings) and empirically identifying two regimes: high-consistency datasets favor max-return modeling, longer context, and global attention; low-consistency datasets favor shorter effective context/local models, and training on advantages reduces variance and improves performance on some low-consistency tasks. Reviewers found the empirical exploration and actionable guidance valuable, but the decision is driven by concerns about (i) the metric’s pipeline dependence and missing sensitivity analysis (clustering/UMAP/DBSCAN choices, hyperparameters), (ii) unclear causal interpretation and confounds (trajectory stitching vs local smoothness, partial observability, stochasticity), (iii) lack of deeper analysis/theory tying dispersion to sequence-model error, and (iv) incomplete experimental reporting (missing parameters, high variance with few seeds, compute overhead for IQL-advantage training, and limited ablations on alternative critics/advantage estimators).

**Reviewer Concerns:**

No author rebuttal is provided in the material above, so the key concerns remain outstanding: sensitivity of the return-consistency pipeline (DBSCAN params, embedding space used, dimensionality reduction), stronger justification of focusing only on high-return clusters, clarifying contradictions/edge cases in conjectures about architecture/context, correcting/clarifying notation issues, providing missing hyperparameters and more seeds with variance reporting, separating trajectory stitching effects from local smoothness (e.g., via learned Q or truncated returns), profiling compute overhead of IQL-advantage training, and ablations showing whether gains generalize beyond IQL advantages (e.g., CQL or other critics) and beyond Reinformer (e.g., DT/QT baselines).

**Reviewer Scores:**

Given no rebuttal/discussion updates shown, I would not expect score changes: jye7 likely 4→4, gUhA 4→4, r5br 2→2, grBi 6→6. Overall sentiment remains mixed with one weak accept and multiple borderline/reject scores anchored by methodological sensitivity and unclear causal interpretation.

---

### Decision · Program_Chairs · 2026-01-26

Reject